# Prevalence and risk factors for asthma, rhinitis, eczema, and atopy among preschool children in an Andean city

Cristina Ochoa-Avilés[1,2]ʘ*, Diana Morillo[1,2]ʘ, Alejandro Rodriguez[2], Philip John Cooper[2,3], Susana Andrade[1], María Molina[1,2], Mayra Parra[1], Andrea Parra-Ullauri[4], Danilo Mejía[1], Alejandra Neira[1,5], Claudia Rodas-Espinoza[5]ʘ, Angélica Ochoa-Avilés[1]ʘ

1 Bioscience Department, Faculty of Chemistry, University of Cuenca, Cuenca, Azuay, Ecuador, 2 School of Medicine, International University of Ecuador, Quito, Ecuador, 3 Institute of Infection and Immunity, St George's University of London, London, United Kingdom, 4 Faculty of Architecture, Arts and Design, Technological University of Indoamérica, Ambato, Tungurahua Ecuador, 5 Research Department, Faculty of Medicine, University of Azuay, Cuenca, Azuay, Ecuador

ʘ These authors contributed equally to this work.
* cristina.ochoaa@ucuenca.edu.ec

**Data Availability Statement:** All relevant data are within the manuscript and its Supporting Information files.

## Abstract

### Background

Limited data are available on prevalence and associated risk factors for atopy and allergic diseases from high-altitude urban settings in Latin America.

### Objective

To estimate the prevalence of atopy, asthma, rhinitis, and eczema, and associations with relevant risk factors in preschool children in the Andean city of Cuenca.

### Methods

A cross-sectional study was undertaken using a representative sample of 535 children aged 3–5 years attending 30 nursery schools in the city of Cuenca, Ecuador. Data on allergic diseases and risk factors were collected by parental questionnaire. Atopy was measured by skin prick test (SPT) reactivity to a panel of relevant aeroallergens. Associations between risk factors and the prevalence of atopy and allergic diseases were estimated using multivariable logistic regression.

### Results

Asthma symptoms were reported for 18% of children, rhinitis for 48%, and eczema for 28%, while SPT reactivity was present in 33%. Population fractions of asthma, rhinitis, and eczema attributable to SPT were 3.4%, 7.9%, and 2.9%, respectively. In multivariable models, an increased risk of asthma was observed among children with a maternal history of rhinitis (OR 1.85); rhinitis was significantly increased in children of high compared to low socioeconomic level (OR 2.09), among children with a maternal history of rhinitis (OR 2.29) or paternal history of eczema (OR 2.07), but reduced among children attending daycare

**Funding:** Funding was provided by CEDIA (Corporación Ecuatoriana para el desarrollo de la Investigación y la Academia), Universidad del Azuay (UDA), Dirección de Investigación, Universidad de Cuenca (DIUC) and Universidad Tecnológica Indoamérica The funders had no role in study design, data collection and analysis, decision to publish, or preparation of the manuscript.

**Competing interests:** The authors have declared that no competing interests exists.

(OR 0.64); eczema was associated with a paternal history of eczema (OR 3.73), and SPT was associated with having a dog inside the house (OR 1.67).

## Conclusions

A high prevalence of asthma, rhinitis, and eczema symptoms were observed among pre-school children in a high-altitude Andean setting. Despite a high prevalence of atopy, only a small fraction of symptoms was associated with atopy. Parental history of allergic diseases was the most consistent risk factor for symptoms in preschool children.

## Introduction

The prevalence of asthma and other allergic diseases (e.g. rhinitis and eczema) has increased over the last twenty years worldwide [1–4]. Atopy is an important risk factor for asthma, rhinitis, and eczema, related to the allergic component of these diseases [5–7]. The International Study of Asthma and Allergies in Childhood (ISAAC) documented high prevalence rates of allergic diseases and atopy in Latin American (LA) countries such as Brazil, Paraguay, Uruguay, Ecuador, and Peru [2,3,8,9]. Allergic diseases are considered to arise through complex interactions between genetic susceptibility and environmental exposures [10], so that temporal trends in prevalence are most likely to be explained by changes in environmental exposure, lifestyle, and living conditions [1]. Among such changes considered to contribute to trends in allergic disease prevalence are climate [11], urbanization [12], air pollution [3], cigarette smoke exposure, breastfeeding, and other behavioral and lifestyles factors [1].

Few published studies have explored risk factors and prevalence of asthma and allergic diseases in preschool children living in urban areas of the high Andes (i.e. >2,500 m), and none have used representative samples in the region [13,14]. Previous cross-sectional studies of allergic disease risk factors were done in tropical and subtropical regions of coastal Ecuador at altitudes below 1,500 m [12,15–20].

The aim of the present study was to describe the prevalence of asthma, rhinitis, and eczema among a representative sample of pre-school children living in the high-altitude Andean city of Cuenca, Ecuador, and identify associated risk factors.

## Materials and methods

### Study design, setting, and sampling

A cross-sectional study was conducted in the city of Cuenca, located in the southern Andean highlands of Ecuador [21] at an altitude of 2,550 meters. The average annual daytime temperature in the city ranges between 15 to 20°C with an average humidity of 84% [22]. The urban area of the city has approximately 332,000 inhabitants of whom 28,603 are aged 3 to 5 years [23], and 90% are mestizos (mixed Spanish–Indigenous ethnicity). The population has on average 11.4 years of schooling [24].

The study involved a cluster random sample of pre-school children aged 3 to 5 years. A sample size of 535 children was required for an estimated atopy prevalence of 20% [16], ±5% precision, and non-participant rate of up to 10%. Thirty preschools or kindergartens were randomly selected with probability proportional to size, stratified by school type (public vs. private) and neighborhood based on a Quality of Life Index (QoL) (high vs. low QoL). QoL characterizes a neighborhood's well-being based on satisfied basic needs (housing

characteristics, basic services, educational level, and access to health services) with each neighborhood classified on a scale of 0 to 2 (where 0 represents complete lack of basic needs, 1 complete coverage, and >1 a quality of life above meeting basic needs [25]. High vs. low QoL scores were defined using the median as cut-off. Eligible preschools for inclusion were: (i) located in urban Cuenca, (ii) attended by children aged 3–5 years, and (iii) having at least 40 such children attending regularly. Within each selected preschool, 40 children were randomly selected from school lists, anticipating an acceptance rate of 50% (i.e. a total of 20 students for each of the 30 schools). The study protocol was approved by the Ethics Committee of the Universidad San Francisco de Quito, Quito, Ecuador (approval 2017-164E), and parents or legal guardians of selected children were asked to give informed written consent.

## Data collection and definitions

Data were collected between June and October 2018 by trained field workers. The ISAAC phase II [26] questionnaire, adapted to local conditions, was administered to the parents or guardian of each child. This questionnaire has been used widely in previous epidemiological studies of children in Ecuador [16–18,20] and collected data on sociodemographic (age, gender) and socio-economic (school type [public/private]) factors as well as on parental occupations, household income, material goods in the household, access to potable water, electricity, and sanitation), and environmental and other relevant risk factors (cat and/or dog inside the house since birth; contacts with animals outside the house; birth order; breastfeeding including duration; attendance at day-care facilities; parental history of allergic diseases; maternal smoking during and after pregnancy, and household exposures to tobacco smoke).

Data on allergic diseases were collected by maternal questionnaire. Here, the term "allergic diseases" is used to refer to symptoms of asthma, rhinitis, eczema, irrespective of the presence of atopy, as widely used in the literature. Asthma was defined as parental reported wheezing in the last 12 months, plus at least one of the following: i) asthma diagnosis ever, ii) wheezing during/after physical exercise in the last 12 months, and iii) sleep interruption due to wheezing in the last 12 months [27]. The presence of nasal congestion or sneezing not associated with a cold in the last 12 months was used to define rhinitis. Eczema was defined as the presence of an itchy rash at any point during the last 12 months involving the folds of the elbows, behind the knees, in front of the ankles, buttocks, or around the neck, ears or eyes [28].

Skin prick testing (SPT) was performed using the following: saline solution as negative control, histamine as positive control, grass mix (*Dactylis glomerata*, *Festuca pratensis*, *Phoa pratensis*, *Phelum pratense*, *Lolium perenne*), tree mix (ash and salix), weed mix (*Plantago*, *Chenopodium*, *Artermisa*, *Ambrosia*, *Parietaria*), fungi (*Alternaria*, *Penicillum*, *Cladosporum*), dust mites (*Dermatophagoides pteronyssinus* and *D.farinae*), tropical mite (*Blomia tropicalis*), dog dander, cat, feather mix (chicken, duck, and goose), cockroach, and latex (INMUNOTEK, Madrid, Spain). Allergens were stored at 4-8ºC and aliquots of antigens were transported to the field on ice packs. Allergens were pricked on the forearm and reaction sizes evaluated after 15 minutes. Reactions were considered positive if the mean wheal size was at least 3 mm greater than the negative saline control [29]. Atopy was defined as a positive reaction to any of the allergens tested.

## Statistical analysis

Data were double entered into Epi Data (EpiData Association, Odense, Denmark). Sociodemographic attributes and risk factors for eczema, rhinitis, and eczema symptoms were reported as percentages. Prevalence of asthma, rhinitis, eczema, and atopy was reported as percentages with 95% confidence intervals (CI) adjusted for sampling using the svy command in

Stata with schools as primary sampling units. Multiple correspondence analysis (MCA) was used to define socioeconomic status (SES) into 3 dimensions or groups (low, medium, and high) using father's/mother's education, father's/mother's occupation, and monthly household income (Fig 1) [18]. Cluster analysis was used to allocate subjects to SES groups identified by MCA. Asthma, rhinitis, and eczema were categorized as atopic vs. non-atopic based on having at least one positive SPT ([12]). Multivariable logistic regression was used to explore associations between allergic diseases and risk factors after controlling for potential confounders. Multivariable models included risk factors with P<0.1 in bivariate analyses after assessment of collinearity using Pearson correlation coefficients. The strength of associations was estimated using odds ratios (OR) with 95% confidence intervals (95% CI) with statistical significance inferred by P<0.05. Population attributable fractions (PAF) were calculated by *PAF = PewX (OR-1)/OR*, where $P_{ew}$ is the prevalence of allergen skin test reactivity among children with the specific symptom of interest as previously described [20]. All analyses were done using Stata V.12.0 (Stata Statistical Software: Release 12. College Station, TX: StataCorp LLC)

## Results

We sampled a total of 535 children attending 30 preschools in the city of Cuenca. Mean age of participants was 4.1 ± 0.7 years (range 3–6 years) and 53.5% were male. Characteristics of the study population and distributions of risk factors are shown in Table 1. A greater proportion belonged to the medium SES (46.2%) than other groups. Most had access to basic services and facilities such as a bathroom (97.7%), electricity (98.7%), and potable water (96.6%). The majority of children were first or second in birth order (82.4%) and had been breastfed (93.8%). A minority had a dog (30.2%) or cat (10.9%) living indoors but did report the presence of animals including dogs and cats outside the home (72.5%). Relatively few parents reported a history of allergic diseases of which rhinitis was the most frequent (mothers 27.2% and fathers 23.5%).

Table 2 shows estimated prevalence of asthma, rhinitis, and eczema symptoms, and sensitization to aeroallergens. Prevalence of asthma, rhinitis, and eczema was 17.8% (95% CI 14.1–21.4), 48.0% (95% CI 43.0–53.2), and 28.0% (95% CI 23.4–32.7), respectively. Positive SPT was observed in 33.5% (95% CI: 29.0–38.0) of children, and domestic mites (*D. pteronyssinus* [21.0%] and *D. farinae* [19.6%]) were the dominant sensitizing allergens.

Stratification of asthma, rhinitis, and eczema by the presence of atopy (measured by a positive allergen skin test) showed non-atopic symptoms to be more prevalent: asthma (atopic 7.8%, 95% CI 4.5–10.8% vs. non-atopic 10%, 95% CI 7.8–12.0%), rhinitis (atopic 19.0%, 95% CI 15–23% vs. non-atopic 29.0%, 95% CI 24.4–33.6%) and eczema (atopic 10.8%, 95% CI 7.8–10.8% vs. non-atopic 17.0%, 95% CI 13.0–21.4%). Only small fractions of symptoms were attributable to atopy (Table 3): asthma (PAF 3.4%), rhinitis (7.9%), and eczema (2.9%)

MCA identified three SES categories: a) low—associated with unemployed parents, basic occupations such as technical and crafts, low income (< = $460) and low parental educational level (<6 years); b) medium—associated with public or private employees, monthly income of $461-$921 and 7–12 years of parental education; and, c) high—represented by professionals, with higher parental educational level (>12 years) and income >$921. The results of multiple correspondence analysis are shown in Fig 1. The dimensions obtained by MCA for SES are shown in Fig 1A. Cluster analysis was used to allocate subjects to each socioeconomic group. Fig 1B shows the spatial distributions of individuals from the cluster analysis, in which dots represent subjects and different colors represent SES groups.

Associations between symptoms of asthma, rhinitis, eczema, and the presence of atopy with potential risk factors are shown in Table 4.k In multivariable analysis, maternal history of

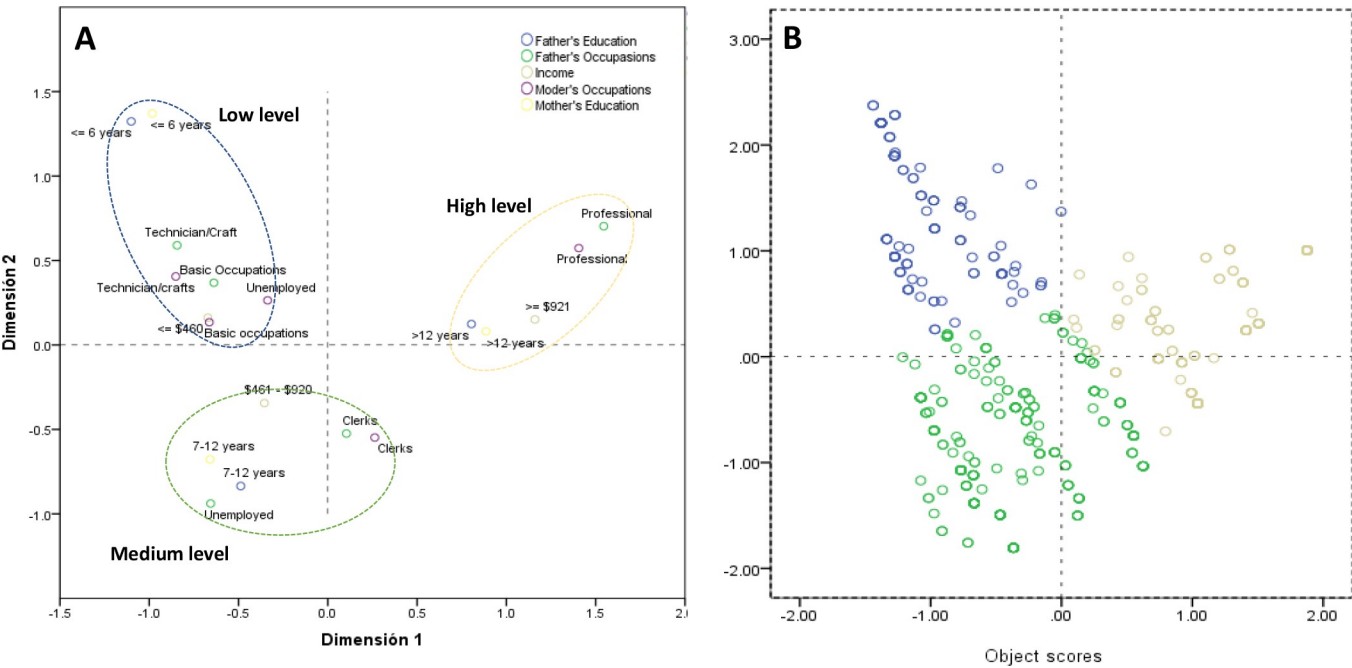

**Fig 1. Multiple correspondence analysis.** A) Spatial distributions of dimensions by socioeconomic groups (low, middle, and high) using the first two dimensions of MCA. B) Cluster analysis using object scores of MCA. Points represent individuals and colors represent socioeconomic groups (blue–low; green -medium; brown–high).

rhinitis was positively associated with asthma and rhinitis symptoms in children. Participants whose mothers had a history of rhinitis were twice as likely to have asthma (OR 1.85, 95% CI 1.0–3.4, P = 0.04) and rhinitis (OR 2.3, 95% CI 1.31–3.98, P = 0.005) symptoms, while those with a paternal history of eczema were four times as likely to have eczema (OR 3.73, 95% CI 1.51–9.20, P = 0.07) and twice as likely to have rhinitis (OR 2.07, 95% CI 1.11–3.86, P = 0.02) symptoms. Daycare attendance was associated with a lower prevalence of rhinitis (OR 0.64, CI 95% 0.46–0.88, P = 0.009). The presence of a dog living inside the house was associated with a greater risk of atopy (OR 1.67, 95% CI 1.05–2.66, P = 0.03). Prevalence of rhinitis symptoms was greater in children of high or medium compared to low SES (high, OR 2.09 [95% CI 1.10–3.96, P = 0.03]; medium, OR 1.75 [95% CI 0.97–3.13, P = 0.06])

## Discussion

To our knowledge, this is the first study to provide an unbiased estimate of the prevalence of atopy and symptoms of asthma, rhinitis, and eczema among preschool children (aged 3–5 years) from a high-altitude setting in Latin America. Our data indicate that despite a high prevalence of symptoms of asthma (17.8%), rhinitis (48.0%), and eczema (28.0%), only a small fraction of these symptoms (i.e. <8%) were attributed to atopy, consistent with findings of previous studies in non-affluent settings in Latin America, including low-attitude tropical regions of Ecuador [3,20,30,31].

Different patterns of risk factors have been identified for atopic and non-atopic asthma. Atopic asthma [12,31–33] has been associated with male sex, previous geohelminth infections, familial history of allergic disease, and respiratory viral infections, while non-atopic asthma has been associated with birth order and sedentary behavior [12,32]. Such observations suggest that atopic and non-atopic asthma may have distinct causal mechanisms and underline the

**Table 1. Characteristics of the study population.**

|  | Frequency | % |
|---|---|---|
| *Sex* | | |
| Male | 286 | 53.5 |
| Female | 249 | 46.5 |
| *Age (years)* | | |
| 3 | 102 | 19.0 |
| 4 | 298 | 55.7 |
| 5 | 125 | 23.4 |
| 6 | 10 | 1.9 |
| *Socioeconomic status* | | |
| Low | 11 | 21.4 |
| Medium | 40 | 46.2 |
| High | 69 | 32.5 |
| *Home services* | | |
| *Bathroom* | 519 | 97.7 |
| *Electricity* | 527 | 98.7 |
| *Potable water (n = 533)* | 515 | 96.6 |
| *Household appliances* | | |
| 2 or less | 84 | 15.7 |
| 3 | 153 | 28.6 |
| 4 | 298 | 55.7 |
| *Environmental Risk Factors* | | |
| *Breastfeeding* | 502 | 93.8 |
| *Breastfeeding (months) (n = 504)* | | |
| < 6 | 88 | 17.5 |
| 6–12 | 189 | 37.5 |
| 13–24 | 167 | 33.1 |
| *Birth order (n = 534)* | | |
| $1^{st}$ -$2^{nd}$ | 440 | 82.4 |
| $> = 3^{rd}$ | 94 | 17.6 |
| *Day-care attendance (n = 532)* | 202 | 38.0 |
| *Mother smoking (n = 531)* | 20 | 3.8 |
| *Family smoking habits (n = 531)* | 92 | 17.3 |
| *Dog inside house (n = 534)* | 161 | 30.2 |
| *Cat inside house (n = 530)* | 58 | 10.9 |
| *Dog outside house* | 359 | 67.6 |
| *Cat outside house* | 144 | 27.2 |
| *Chicken outside house (n = 530)* | 97 | 18.3 |
| *Pig outside house (n = 530)* | 17 | 3.2 |
| *Any animal outside house* | 488 | 72.5 |
| *Contact with animals in farms (n = 530)* | 159 | 30.0 |
| *Parental history of allergic disorders* | | |
| *Maternal asthma (n = 528)* | 24 | 4.6 |
| *Maternal rhinitis (n = 515)* | 140 | 27.2 |
| *Maternal eczema (n = 499)* | 65 | 13.0 |
| *Paternal asthma (n = 484)* | 13 | 2.7 |
| *Paternal rhinitis (n = 472)* | 111 | 23.5 |
| *Paternal eczema (n = 463)* | 37 | 8.0 |

**Table 2. Prevalence of atopy (measured by aeroallergen skin prick testing [SPT]) and symptoms of asthma, rhinitis, eczema and by atopic status.**

| Allergic diseases | % (95% CI) |
|---|---|
| Atopy | 33.5 (29.0–38.0) |
| Asthma | 17.8(14.1–21.4) |
| Atopic asthma | 7.8 (4.5–10.8) |
| Non atopic asthma | 10.0 (7.8–12.0) |
| Rhinitis | 48.0 (43.0–53.2) |
| Atopic rhinitis | 19.0 (15.0–23.1) |
| Non atopic rhinitis | 29.0 (24.4–33.6) |
| Eczema | 28.0 (23.4–32.7) |
| Atopic eczema | 10.8 (7.8–13.8) |
| Non atopic eczema | 17.2 (13.0–21.4) |
| **Sensitization to aeroallergens by SPT** | |
| Mites | 24.3 (19.7–28.8) |
| D. farinae | 21.0 (16.5–25.0) |
| D. pteronyssinus | 19.6 (15.4–24.0) |
| B. tropicalis | 4.3 (1.9–6.7) |
| Pollen* | 3.7 (2.3–5.1) |
| Cockroach | 2.6 (1.1–4.1) |
| Cat | 2.1 (0.4–3.6) |
| Dog| | 1.5 (0.4–2.5) |
| Salix | 1.5 (0.04–2.6) |
| Feather mix*** | 1.1 (0.1–2.1) |
| Ash tree | 0.9 (0.009–1.9) |
| Fungi** | 0.9 (0.003–2.0) |
| Latex | 0.7 (0.02–1.4) |

* plantago, *Chenopodium*, mugwort, ragweed, *Parietaria*

** *Alternaria*, *Cladosporium*, *Penicillium*

*** chicken, goose, duck

need for more research in Latin American settings [34]. The strength of the association between atopy and asthma tends to increase with age: atopic asthma tends to be associated with a more persistent disease likely to continue into adulthood [12,35–37]. Longitudinal studies have provided evidence of three different phenotypes of asthma in childhood: 1) transient asthma associated with viral respiratory tract infections that often resolves by 6 years of age, 2) non-atopic asthma associated with viral respiratory tract infections that may persist beyond 6 years but which resolves by adulthood, and, 3) atopic asthma that persists into adulthood accompanied by a more severe clinical course [12].

Previous studies have identified genetic, environmental, behavioral, and socioeconomic factors associated with the development of allergic diseases in childhood, likely to reflect complex interactions between genes and environmental exposures [38], [39,40]. Genetic studies have identified polymorphisms associated with both atopic and non-atopic asthma [27,41,42], highlighting the importance of family history as a risk factor. Bjerg. et al. [43] reported parental asthma as a risk factor for childhood asthma, and data from Europe showed that family history of rhinitis was associated with a four-fold increased risk of developing asthma and two to six-fold increased risk of developing rhinitis [44,45]. In our study, family history of allergic disorders was the most consistent factor associated with symptoms of asthma, rhinitis, and eczema.

**Table 3. Fractions of symptoms of asthma, rhinitis, eczema attributable to atopy.**

| Disease | Prevalence of allergen skin prick test reactivity (%) | Adjusted OR* | Population attributable fraction (%) |
|---|---|---|---|
| Asthma | 7.9% | 1.75 | 3.4 |
| Rhinitis | 19.1% | 1.71 | 7.9 |
| Eczema | 10.8% | 1.36 | 2.9 |

*OR for association between symptoms and atopy

The potential role of epigenetic changes representing the role of gene-environment interactions in the development of asthma has been explored also. Yang and colleagues [40] identified 81 regions on the genome that were differentially methylated in asthmatic children aged between 6 and 12 years. Further studies to identify polymorphisms and epigenetic alterations associated with allergic diseases, as well as variants in genes associated with reduced pharmacologic response, are required for a better understanding of asthma and its treatment among Andean children with asthma [41,46–49].

Among environmental factors, helminth infections in early life [50,51], exposure to dust mites, cockroach, cigarette smoke [38], and the presence of pets and/or farm animals during childhood have all strong effects on allergic diseases [3,11,50,52]. Most previous studies from Ecuador have identified factors related to poverty and dirt (lack of access to potable water, migration, home infrastructure, household pets, contact with farm animals, and socioeconomic level) as risk factors for asthma in poor tropical populations [16–18,20]. In the present study, environmental and socioeconomic factors were most likely to affect the prevalence of rhinitis symptoms: daycare attendance was associated with a lower prevalence of rhinitis. Attendance at daycare facilities is associated with a greater contact among young children and much greater exposure to infections during childhood in comparison with children who stay at home [51,53–55]. There is some evidence that exposure to infections may affect allergic rhinitis [53]; helminth infections have been associated with a lower incidence of allergic rhinitis [53,54,56,57]. Higher socioeconomic status was associated with increased prevalence of rhinitis as indicated by previous studies [50,58–60]. The only factor associated with atopy in the present study was having a dog living in the house at any time since the child's birth. Although previous studies have reported dog ownership during pregnancy and first year of life as protective against allergic diseases [61], associations with atopy have been less clear. Studies in Europe have reported low levels of mite allergens in high altitude settings (>1500 m) related to low humidity [62,63]. Previous studies in Quito, another high altitude city (2,800 m altitude) in the Ecuadorian Andes, showed significant levels of indoor dust mites (*D. pteronyssinus and D. farinae*) [64] and mite allergens associated with respiratory allergy [65]. Data from this study indicate that mites are important sensitizing allergens in pre-school children in another Andean urban setting of Ecuador with a relatively high humidity (average 83.7%, range 77.5–89.4%) [66].

Research in pre-school children (3–5 years) in China using the ISAAC questionnaire reported similar prevalence to that observed in the present study: asthma (14%), rhinitis (40%), and eczema (21%) [67]. Although we were unable to identify studies of representative samples of young children elsewhere in Latin America, our estimate of rhinitis was higher than that reported previously among children aged 1 to 4 years in 6 Colombian cities (32%) [68] while eczema prevalence (28%) was similar to that reported among schoolchildren in Bogotá, Colombia (25%) [69] and 6 to 7 year-olds in ISAAC (22%) [70]. Cuenca and Bogotá are Andean cities with high altitude near the Equator with low mean annual temperatures (range 9 to 21ºC): it has been suggested that eczema prevalence may be inversely associated

**Table 4. Risk factors associated with atopy and symptoms of asthma, rhinitis, and eczema.**

| Predictors | Asthma (N = 95/535) | | | Rhinitis (N = 257/535) | | | Eczema (N = 150/535) | | | Atopy (N = 179/535) | | |
|---|---|---|---|---|---|---|---|---|---|---|---|---|
| | Frequency N (%) | Bivariate OR (95% CI) P Value | Multivariable OR (95% CI) P Value | Frequency N (%) | Bivariate OR (95% CI) P Value | Multivariable OR (95% CI) P Value | Frequency N (%) | Bivariate OR (95% CI) P Value | Multivariable OR (95% CI) P Value | Frequency N (%) | Bivariate OR (95% CI) P Value | Multivariable OR (95% CI) P Value |
| *Socioeconomic status* | | | | | | | | | | | | |
| Low | 13 (13.7) | 1 | | 35 (13.6) | 1 | | 33 (22.0) | 1 | | 29 (16.2) | 1 | |
| Medium | 47 (49.5) | 1.83 (1.08–3.12) 0.03 | 1.42 (0.71–2.83) 0.31 | 114 (44.4) | 1.96 (1.22–3.15) 0.07 | **1.75 (0.97–3.13) 0.06** | 80 (53.3) | 1.18 (0.69–2.00) 0.52 | | 88 (49.2) | 1.63 (1.09–2.45) 0.02 | 1.17 (0.92–1.48) 0.20 |
| High | 32 (33.7) | 1.76 (0.91–3.37) 0.08 | 0.90 (0.38–2.10) 0.80 | 99 (38.5) | 3.07 (1.79–5.25) <0.001 | **2.09 (1.10–3.96) 0.03** | 33 (22.0) | 0.57 (0.28–1.14) 0.10 | | 58 (32.4) | 1.47 (0.88–2.47) 0.13 | 1.47 (0.88–2.47) 0.13 |
| *Type of school* | | | | | | | | | | | | |
| Public | 43 (45.3) | | | 130 (50.6) | | | 92 (61.3) | | | 88 (49.2) | | |
| Private | 52 (54.7) | 1.56 (1.00–2.45) 0.05 | | 127 (49.4) | 1.34 (0.91–1.98) 0.13 | | 58 (38.6) | 0.67 (0.43–1.07) 0.09 | 0.80 (0.43–1.48) 0.47 | 91 (50.8) | 1.37 (0.96–1.97) 0.08 | 1.43 (0.96–2.13) 0.07 |
| *Bathroom* | | | | | | | | | | | | |
| Hygienic service | 94 (99.0) | | | 251 (97.7) | | | 148 (98.6) | | | 171 (95.5) | | |
| Latrine or field | 1.0 (1.1) | 0.41 (0.07–2.58) 0.33 | | 4 (1.6) | 0.53 (0.24–1.21) 0.13 | | 1 (0.6) | 0.23 (0.04–1.34) 0.10 | 0.49 (0.11–2.11) 0.34 | 6 (3.3) | 2.04 (0.54–7.61) 0.29 | |
| *Household appliances* | | | | | | | | | | | | |
| 2 or less | 18 (19.0) | | | 47 (14.4) | | | 28 (18.6) | | | 32 (17.9) | | |
| 3 or more | 77 (81.1) | 0.93 (0.67–1.29) 0.67 | | 220 (85.7) | 1.30 (0.97–1.74) 0.078 | 1.04 (0.75–1.44) 0.79 | 122 (81.2) | 0.78 (0.6–1.00) 0.05 | 0.97 (0.67–1.41) 0.89 | 147 (82.1) | 0.92 (0.74–1.15) 0.46 | |
| *Birth order* | | | | | | | | | | | | |
| 1–2 | 18 (19.0) | | | 45 (17.5) | | | 31 (20.6) | | | 33 (18.4) | | |
| 3 or more | 77 (81.1) | 0.90 (0.47–1.69) 0.73 | | 212 (82.5) | 1.01 (0.67–1.53) 0.95 | | 118 (78.6) | 0.74 (0.52–1.06) 0.10 | 0.74 (0.43–1.29) 0.28 | 145 (81.0) | 0.91 (0.57–1.45) 0.69 | |
| *Breastfeeding (months)* | | | | | | | | | | | | |
| < 6 | 14 (14.7) | | | 47 (18.3) | | | 22 (14.6) | | | 30 (16.8) | | |
| 6–12 | 30 (31.6) | | | 92 (35.8) | | | 48 (32.0) | | | 63 (35.2) | | |
| 12–24 | 33 (34.7) | | | 71 (27.6) | | | 47 (31.3) | | | 60 (33.5) | | |
| >24 | 14 (14.7) | 1.19 (0.94–1.51) 0.14 | | 30 (11.7) | 0.90 (0.76–1.07) 0.22 | | 24 (16.0) | 1.22 (1.01–1.49) 0.04 | 1.05 (0.79–1.39) 0.72 | 17 (9.5) | 0.96 (0.78–1.20) 0.74 | |
| *Daycare* | | | | | | | | | | | | |
| Yes | 35 (36.8) | | | 119 (46.3) | | | 49 (32.6) | | | 63 (35.2) | | |
| No | 60 (63.2) | 1.06 (0.68–1.65) 0.78 | | 136 (52.9) | 0.49 (0.37–0.64) 0.001 | **0.64 (0.46–0.88) 0.009** | 101 (67.3) | 1.38 (0.88–2.16) 0.16 | | 114 (63.7) | 1.16 (0.77–1.77) 0.48 | |
| *Mother smoking* | | | | | | | | | | | | |
| Yes | 2 (2.1) | | | 11 (4.3) | | | 6 (4.0) | | | 3 (1.7) | | |
| No | 93 (97.9) | 0.50 (0.14–1.78) 0.27 | | 244 (94.9) | 1.34 (0.61–2.92) 0.47 | | 142 (94.6) | 1.11 (0.39–3.22) 0.84 | | 133 (96.6) | 0.34 (0.12–0.96) 0.04 | 0.39 (0.11–1.34) 0.13 |
| *Family smoking* | | | | | | | | | | | | |
| Yes | 19 (21.0) | | | 47 (18.3) | | | 34 (22.6) | | | 22 (12.3) | | |

*(Continued)*

**Table 4.** (Continued)

| Predictors | Asthma (N = 95/535) | | | Rhinitis (N = 257/535) | | | Eczema (N = 150/535) | | | Atopy (N = 179/535) | | |
|---|---|---|---|---|---|---|---|---|---|---|---|---|
| | Frequency | Bivariate | Multivariable | Frequency | Bivariate | Multivariable | Frequency | Bivariate | Multivariable | Frequency | Bivariate | Multivariable |
| | N (%) | OR (95% CI) P Value | OR (95% CI) P Value | N (%) | OR (95% CI) P Value | OR (95% CI) P Value | N (%) | OR (95% CI) P Value | OR (95% CI) P Value | N (%) | OR (95% CI) P Value | OR (95% CI) P Value |
| *No* | 75 (79.0) | 1.26 (0.65–2.45) 0.47 | | 207 (80.5) | 1.17 (0.69–2.00) 0.56 | | 115 (76.6) | 1.65 (1.12–2.44) 0.01 | 1.30 (0.75–2.24) 0.32 | 156 (87.2) | 0.57 (0.36–0.90) 0.01 | 0.67 (0.38–1.16) 0.15 |
| *Dog (inside house)* | | | | | | | | | | | | |
| *No* | 63 (66.3) | | | 185 (71.9) | | | 103 (68.6) | | | 137 (76.5) | | |
| *Yes* | 32 (33.7) | 0.82 (0.49–1.37) 0.43 | | 72 (28.0) | 1.22 (0.82–1.80) 0.33 | | 47 (31.3) | 0.93 (0.62–1.38) 0.70 | | 42 (23.5) | **1.64 (1.11–2.44) 0.01** | **1.67 (1.05–2.66) 0.03** |
| *Maternal rhinitis* | | | | | | | | | | | | |
| *No* | 38 (56.8) | | | 154 (60.0) | | | 43 (28.6) | | | 125 (69.8) | | |
| *Yes* | 54 (40.0) | **2.21 (1.38–3.55) 0.002** | **1.85 (1.01–3.41) 0.04** | 92 (35.8) | **2.75 (1.79–4.22) 0.001** | **2.29 (1.31–3.98) 0,005** | 98 (28.6) | 1.25 (0.78–2.01) 0.35 | | 46 (25.8) | 0.98 (0.67–1.42) 0.91 | |
| *Maternal eczema* | | | | | | | | | | | | |
| *No* | 18 (18.9) | | | 201 (78.2) | | | 108 (72.0) | | | 145 (81.0) | | |
| *Yes* | 69 (72.3) | 2.03 (0.96–4.28) 0.064 | 1.58 (0.68–3.79) 0.27 | 38 (14.8) | 1.63 (0.84–3.17) 0.15 | | 28 (18.6) | **2.28 (1.42–3.68) 0.001** | 1.73 (0.91–3.28) 0.08 | 25 (13.9) | 1.25 (0.72–2.15) 0.43 | |
| *Paternal rhinitis* | | | | | | | | | | | | |
| *No* | 21 (22.1) | | | 158 (61.5) | | | 90 (60.0) | | | 114 (63.7) | | |
| *Yes* | 58 (61.1) | 1.22 (0.68–2.17) 0.49 | | 68 (26.5) | **2.03 (1.34–3.07) 0.001** | 1.16 (0.67–2.00) 0.58 | 35 (23.3) | 1.39 (0.88–2.19) 0.16 | | 39 (21.8) | 1.17 (0.75–1.84) 0.48 | |
| *Paternal eczema* | | | | | | | | | | | | |
| *No* | 68 (71.6) | | | 200 (77.8) | | | 101 (67.3) | | | 143 (80.0) | | |
| *Yes* | 9 (9.5) | 1.69 (0.76–3.75) 0.186 | | 25 (9.7) | **2.35 (1.30–4.26) 0.005** | **2.07 (1.11–3.86) 0.02** | 21 (14.0) | **4.22 (2.30–7.75) <0.001** | 3.73 (1.51–9.20) 0.07 | 11 (6.1) | 0.84 (0.38–1.84) 0.65 | |

with temperature through differences in sun exposure [11]. The estimated prevalence of eczema in children in ISAAC was higher in Latin America than that reported from temperate countries [71]. Recent analyses comparing ISAAC with other definitions of eczema showed a tendency for the ISAAC definition to overestimate eczema prevalence in tropical compared to sub-tropical settings [72], largely because of conditions (e.g. miliaria and arthropod bites) whose clinical presentations may inflate ISAAC-based estimates. It should be emphasized, however, that in the subtropical climate of Cuenca, such conditions are less common. Previous studies in Ecuador showed a lower prevalence of asthma and rhinitis compared to our results [15,18]. Differences in asthma and rhinitis prevalence could be related to the fact that previous studies in Ecuador have been conducted in older children living in humid rural tropical communities at sea level [15,19,73]. Aside from the population characteristics, urbanization, altitude [74], temperature, humidity, [11,75–77], and regional differences could also be linked to variations in prevalence [26].

A limitation of the present study was the cross-sectional design that does not allow us to determine the direction of causality between potential risk factors and disease outcomes. However, the representative sample used in the study allowed us to minimize potential biases in estimating prevalence and our findings are likely to be relevant to young children living in Andean cities. Because allergen extracts of pollen from representative plants are not commercially available for the Andean region, we used pollen extracts from European plants in the present study; this could have underestimated allergic sensitization to plant pollens in our study population.

## Conclusion

In the present study, we observed a high prevalence of asthma, rhinitis, eczema symptoms among a representative sample of preschool children living in a high-altitude Andean city in Ecuador. Despite a high prevalence of atopy, predominantly to domestic mites, only a small proportion of 'allergic' disease symptoms (<8%) were attributable to atopy. Parental history of allergic diseases was the most consistent risk factor for symptoms in these young children, indicating the importance of genetic susceptibility, while few of the standard environmental exposures measured by questionnaire were associated with symptoms. Future studies should examine a wider range of environmental exposures relating to urbanization such as the role of indoor and outdoor pollution.

## Supporting information

**S1 Data.**
(XLS)

## Acknowledgments

We are grateful for the participation and support of students, parents, school directors, and teachers in the study. Additionally, we acknowledge the contribution of staff of CEDIA, University of Azuay, the Bioscience Department of the University of Cuenca and to Elizabeth Rodas for language editing. This work was developed within a joint postgraduate program of Vlir Network Ecuador.

## Author Contributions

**Conceptualization:** Susana Andrade, Danilo Mejía, Claudia Rodas-Espinoza, Angélica Ochoa-Avilés.

**Data curation:** Cristina Ochoa-Avilés, Diana Morillo, Alejandro Rodriguez, María Molina, Andrea Parra-Ullauri.

**Formal analysis:** Cristina Ochoa-Avilés, Diana Morillo, Alejandro Rodriguez, Danilo Mejía.

**Funding acquisition:** Susana Andrade, Mayra Parra, Andrea Parra-Ullauri, Danilo Mejía, Claudia Rodas-Espinoza, Angélica Ochoa-Avilés.

**Investigation:** Cristina Ochoa-Avilés, Diana Morillo, Susana Andrade, Andrea Parra-Ullauri, Claudia Rodas-Espinoza, Angélica Ochoa-Avilés.

**Methodology:** Cristina Ochoa-Avilés, Diana Morillo, Philip John Cooper, María Molina, Mayra Parra, Andrea Parra-Ullauri, Claudia Rodas-Espinoza, Angélica Ochoa-Avilés.

**Project administration:** Susana Andrade, Mayra Parra, Claudia Rodas-Espinoza.

**Resources:** Susana Andrade, Claudia Rodas-Espinoza, Angélica Ochoa-Avilés.

**Supervision:** Susana Andrade, Mayra Parra, Danilo Mejía, Alejandra Neira, Claudia Rodas-Espinoza, Angélica Ochoa-Avilés.

**Validation:** Cristina Ochoa-Avilés, Diana Morillo, María Molina.

**Visualization:** Cristina Ochoa-Avilés, Philip John Cooper, Andrea Parra-Ullauri.

**Writing – original draft:** Cristina Ochoa-Avilés.

**Writing – review & editing:** Cristina Ochoa-Avilés, Diana Morillo, Alejandro Rodriguez, Philip John Cooper, Susana Andrade, María Molina, Alejandra Neira, Claudia Rodas-Espinoza, Angélica Ochoa-Avilés.

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
