## [Decision Letter · Decision Letter 0]

20 Feb 2020

PONE-D-20-02044

Prevalence and risk factors for asthma, rhinitis, eczema and atopy among preschool children in an Andean city.

PLOS ONE

Dear Mrs Ochoa,

Thank you for submitting your manuscript to PLOS ONE. After careful consideration, we feel that it has merit but does not fully meet PLOS ONE’s publication criteria as it currently stands. Therefore, we invite you to submit a revised version of the manuscript that addresses the points raised during the review process.

Although this is an interesting epidemiological study adding also new points of view on risk factors for asthma, rhinitis, eczema and atopy, there are several points which need further elaboration in the paper: definitions of "eczema" or "atopy" are not clearly defined throughout the paper, skin prickt test issue raised by a reviewer, a deeper discussion regarding the association of high-altitude and atopy should be provided (if any), as well as some issues regarding the logistic regression analysis should be addressed

We would appreciate receiving your revised manuscript by Apr 05 2020 11:59PM. To enhance the reproducibility of your results, we recommend that if applicable you deposit your laboratory protocols in protocols.io, where a protocol can be assigned its own identifier (DOI) such that it can be cited independently in the future. For instructions see: http://journals.plos.org/plosone/s/submission-guidelines#loc-laboratory-protocols

We look forward to receiving your revised manuscript.

Kind regards,

Claudio Andaloro

Academic Editor

PLOS ONE

Journal Requirements:

Reviewers' comments:

Reviewer's Responses to Questions

**Comments to the Author**

1. Is the manuscript technically sound, and do the data support the conclusions?

Reviewer #1: Yes

Reviewer #2: Partly

2. Has the statistical analysis been performed appropriately and rigorously? 

Reviewer #1: Yes

Reviewer #2: Yes

3. Have the authors made all data underlying the findings in their manuscript fully available?

Reviewer #1: No

Reviewer #2: No

4. Is the manuscript presented in an intelligible fashion and written in standard English?

Reviewer #1: Yes

Reviewer #2: Yes

5. Review Comments to the Author

Reviewer #1: This is an interesting epidemiological study concerning prevalence of Eczema, Asthma and Rhinitis in a Highland pediatric Population of the Andes which seems to be a new Patient collective. Also, it adds some Information, whether higher social income is also a Risk factor in developping countries in the same way as in the industrialized world.

Although the study is well written with a highly sophisticated Level of statistical work-up, I fave some very Basic Major remarks mainly concerning the definitions used in this paper.

The high prevalence of eczema patients is astounding and makes me suspicious that the term "eczema" is too vaguely defined in a pediatric Population especially prone to viral infections with a high Risk of developping infection-connected eczema rather than true atopic eczema.

Also the term "atopy" is not clearly defined throughout the paper: how do the authors use this term? An atopic parent?, A positive Skin prick test?

Minor remark: Is the Skin prickt test series truely representative for the plants abundant in the Highland Andes? They rather seem to be fit for a central European Population completely ignoring endemic plant species in the Highland Ecuador and more General Altiplano Region. Also Information regarding the manufacturer of the test substances is missing.

This was a field study. How did the authors guarantee that allergenicity was not lost by using Skin prick test Solutions that were not stored properly betwenn 4 and 8 Degree Celsius. Especially house dust mite extracts will soon loose their activity through the proteolitic activity of the Major house dust mite allergen components when stored at room temperature for longer time.

Methods: The authors describe "Scratch test". Hopefully this is only a wrong description and the authors used "Skin prick Tests"?

Reviewer #2: 1) Please elaborate more about the rationale about high attitude level (higher than 2,500m) and the atopy.

2) Regarding the logistic regression analysis, the value of each variable should be display also. By showing only the percentage and Odds ratio are not enough for determining the degree of association of risk factors.

6. PLOS authors have the option to publish the peer review history of their article (what does this mean?). If published, this will include your full peer review and any attached files.

Reviewer #1: Yes: Stefan Wöhrl, Floridsdorf Allergy Center (FAZ), Austria, Europe

Reviewer #2: No

---

## [Author Response · Author response to Decision Letter 0]

25 Mar 2020

-The high prevalence of eczema patients is astounding and makes me suspicious that the term "eczema" is too vaguely defined in a pediatric Population especially prone to viral infections with a high Risk of developing infection-connected eczema rather than true atopic eczema.

R: We thank the reviewer for this insightful comment. We did observe a high prevalence of eczema in our pediatric population of children aged 3 to 5 years (28%). We used the ISAAC definition for eczema, that has been used widely in epidemiological studies in children and estimated a prevalence similar to that described in other epidemiological studies in Latin America using the same definition. For example, a similar prevalence (22.5%) was estimated among Ecuadorian children aged 6-7 years in the ISAAC Phase III study (Odhiambo et al. 2009). The differential diagnosis of eczema in 3-5-year olds in this setting include seborrheic dermatitis, scabies and other arthropod bites, miliaria, fungal infections, etc. Recent analyses comparing ISAAC with other definitions of eczema have shown a tendency of the ISAAC definition to overestimate eczema prevalence in tropical compared to sub-tropical settings (Sánchez, Sánchez, and Cardona 2018). It should be emphasized that the City of Cuenca is sub-tropical where conditions that might inflate ISAAC-based prevalence estimates, such as miliaria and scabies, are less common. We have added now text explaining this to the Discussion (page 14). 

 -Also, the term "atopy" is not clearly defined throughout the paper: how do the authors use this term? An atopic parent? A positive Skin prick test?

R: We thank the reviewers comment, the term atopy was defined as “a positive [skin test] reaction to any of the allergens tested” on page 6 of the manuscript following the definition of the European Academy of allergen and clinical immunology (EAACI)(Johansson et al. 2004). The definition of atopy is provided also in the abstract: “Atopy was measured by skin prick test (SPT) reactivity to a panel of relevant aeroallergens”. We have clarified the definition of atopy used in this study in the relevant section of the Results on page 9. We acknowledge that the words ‘allergic’ and ‘atopic’ are often used interchangeably so have clarified that ‘allergic diseases’ refers to ‘symptoms of the commonly called ‘allergic diseases”, asthma, rhinitis, eczema, without determination of the actual presence of atopy’ (page 6). 

-Minor remark: Is the Skin prick test series truly representative for the plants abundant in the Highland Andes? They rather seem to be fit for a central European Population completely ignoring endemic plant species in the Highland Ecuador and more General Altiplano Region. Also Information regarding the manufacturer of the test substances is missing.

R: The plant species used for skin prick testing in this study were based on clinical experience and availability of purified extracts. Commercially available extracts representative of the plant species in highland Ecuador or the Andean region are not available. We have added a statement to the limitation paragraph on page 15 discussing this limitation. Information on commercial source of allergens manufacturer has been added to methods (page 6). 

-This was a field study. How did the authors guarantee that allergenicity was not lost by using Skin prick test Solutions that were not stored properly between 4 and 8 Degree Celsius. Especially house dust mite extracts will soon lose their activity through the proteolytic activity of the Major house dust mite allergen components when stored at room temperature for longer time.

R: Allergens were stored at 4-8ºC and aliquots of antigens were transported to the field on ice packs. This sentence has been added to the text on page:

-Methods: The authors describe "Scratch test". Hopefully this is only a wrong description and the authors used "Skin prick Tests"?

R: We thank the reviewer for highlighting this error in the text - we did skin prick tests. The word ‘scratch’ has been replaced by ‘prick’. Text now reads: “Allergens were pricked on the forearm and reaction sizes evaluated after 15 minutes” (page 6). 

-Please elaborate more about the rationale about high attitude level (higher than 2,500m) and the atopy.

Text about atopy and high alttitude level has been added to the manuscript.

-Regarding the logistic regression analysis, the value of each variable should be display also. By showing only the percentage and Odds ratio are not enough for determining the degree of association of risk factors.

The value of each value has been added to the fig. 4

---

## [Decision Letter · Decision Letter 1]

20 Apr 2020

PONE-D-20-02044R1

Prevalence and risk factors for asthma, rhinitis, eczema and atopy among preschool children in an Andean city.

PLOS ONE

Dear Mrs Ochoa,

Thank you for submitting your manuscript to PLOS ONE. After careful consideration, we feel that it has merit but does not fully meet PLOS ONE’s publication criteria as it currently stands. Therefore, we invite you to submit a revised version of the manuscript that addresses the points raised during the review process.

You have provided sufficient efforts to address previous peer review report, but the manuscript needs some minor

adjustments. You need to check for grammatical and typographical errors in the text.

We would appreciate receiving your revised manuscript by Jun 04 2020 11:59PM. To enhance the reproducibility of your results, we recommend that if applicable you deposit your laboratory protocols in protocols.io, where a protocol can be assigned its own identifier (DOI) such that it can be cited independently in the future. For instructions see: http://journals.plos.org/plosone/s/submission-guidelines#loc-laboratory-protocols

We look forward to receiving your revised manuscript.

Kind regards,

Claudio Andaloro

Academic Editor

PLOS ONE

Reviewers' comments:

Reviewer's Responses to Questions

**Comments to the Author**

1. If the authors have adequately addressed your comments raised in a previous round of review and you feel that this manuscript is now acceptable for publication, you may indicate that here to bypass the “Comments to the Author” section, enter your conflict of interest statement in the “Confidential to Editor” section, and submit your "Accept" recommendation.

Reviewer #1: All comments have been addressed

Reviewer #2: (No Response)

Reviewer #3: All comments have been addressed

2. Is the manuscript technically sound, and do the data support the conclusions?

Reviewer #1: Yes

Reviewer #2: No

Reviewer #3: Yes

3. Has the statistical analysis been performed appropriately and rigorously? 

Reviewer #1: Yes

Reviewer #2: No

Reviewer #3: Yes

4. Have the authors made all data underlying the findings in their manuscript fully available?

Reviewer #1: Yes

Reviewer #2: No

Reviewer #3: Yes

5. Is the manuscript presented in an intelligible fashion and written in standard English?

Reviewer #1: Yes

Reviewer #2: No

Reviewer #3: No

6. Review Comments to the Author

Reviewer #1: (No Response)

Reviewer #2: (No Response)

Reviewer #3: There is an error in the fourth sentence in the discussion section. After "and eczema (28.0%)" you should have a comma. Please check carefully for other errors.

7. PLOS authors have the option to publish the peer review history of their article (what does this mean?). If published, this will include your full peer review and any attached files.

Reviewer #1: Yes: Stefan Wöhrl

Reviewer #2: No

Reviewer #3: No

---

## [Author Response · Author response to Decision Letter 1]

26 May 2020

Rebuttal Letter Article: [PONE-D-20-02044]-[EMID:81850e4da8ff6666]

Title: Prevalence and risk factors for asthma, rhinitis, eczema, and atopy among preschool children in an Andean city.

We appreciate the valuable comments. This document is written as follows: reviewer’s

comments are in italic and bold, responses to the comments are preceded by the letter R; extracts of the manuscript are in blue.

Reviewer's Report:

There is an error in the fourth sentence in the discussion section. After "and eczema (28.0%)" you should have a comma. Please check carefully for other errors.

R: we appreciate the careful revision that has been made to the manuscript. We corrected the error in the fourth sentence of the discussion.

“… symptoms of asthma (17.8%), rhinitis (48.0%), and eczema (28.0%), only a small fraction of…”

You need to check for grammatical and typographical errors in the text.

R: grammatical and typographical errors were carefully checked trough the manuscript. All the modifications made were highlighted in the text in blue.

---

## [Editor Report · Decision Letter 2]

1 Jun 2020

Prevalence and risk factors for asthma, rhinitis, eczema, and atopy among preschool children in an Andean city.

PONE-D-20-02044R2

Dear Dr. Ochoa,

We are pleased to inform you that your manuscript has been judged scientifically suitable for publication and will be formally accepted for publication once it complies with all outstanding technical requirements.

With kind regards,

Claudio Andaloro

Academic Editor

PLOS ONE
---

## [Editor Report · Acceptance letter]

10 Jun 2020

PONE-D-20-02044R2 

Prevalence and risk factors for asthma, rhinitis, eczema, and atopy among preschool children in an Andean city. 

Dear Dr. Ochoa-Avilés:

I'm pleased to inform you that your manuscript has been deemed suitable for publication in PLOS ONE. Congratulations! Your manuscript is now with our production department. 

Kind regards, 

on behalf of

Dr. Claudio Andaloro 

Academic Editor

PLOS ONE